# Fragmentation of hunting bullets observed with synchrotron radiation: Lighting up the source of a lesser-known lead exposure pathway

**Adam F. G. Leontowich**[1]*, **Arash Panahifar**[1,2], **Ryan Ostrowski**[3]

**1** Canadian Light Source Inc., Saskatoon, Saskatchewan, Canada, **2** Department of Medical Imaging, University of Saskatchewan College of Medicine, Saskatoon, Saskatchewan, Canada, **3** Royal University Hospital, Saskatoon, Saskatchewan, Canada

* adam.leontowich@lightsource.ca

## Abstract

Bullet fragments have been previously observed in the remains and edible portions of big game animals that were harvested using rifles. The fragmentation issue has attracted attention because traditional hunting bullets are more than 70% lead, which is toxic to humans and scavengers in the ecosystem. We prepared gunshot wounds in ballistic gelatin blocks, and then applied synchrotron X-ray imaging technology to the bullet fragmentation process for the first time. The K edge subtraction (KES) imaging method allowed a clear separation of lead in an image from false positives, including the other major bullet component, copper, and non-lead objects such as bone fragments. The superior brightness of synchrotron radiation was also harnessed to resolve thousands of embedded sub-10 µm fragments, a size range not previously observed using commonly applied X-ray imaging modalities. The results challenge the current understanding of the maximum extent that fragments may be distributed, and the effectiveness of imaging methods used to screen wild game donations at food banks for lead bullet fragments.

## Introduction

Canada has large and extensive populations of "big game" animals, including several species of deer, moose, elk, pronghorn, and bear. For millennia, Indigenous peoples both revered and depended on these animals for food, clothing, shelter, tools, and ceremonial purposes. Over the last 400 years, the North American continent has seen massive immigration, extensive agriculture, growth of urban centers and trade networks. Basic survival is no longer as dependent on hunting. But hunting continues to have an important place in Canadian culture and community, especially among Indigenous peoples where wild game remains a significant component of the diet.

Big game hunting practices have evolved over time with improvements in technology, from the buffalo jump which required the participation and coordination of whole communities, to bows and arrows, the (re)introduction of horses, firearms, automobiles and aircraft. Hunting

**Data Availability Statement:** All raw data has been uploaded to the Canadian Federated Research Data Repository. Accession numbers and/or DOIs will be made available after acceptance.

**Funding:** The authors received no specific funding for this work. Part of the research described in this paper was performed at the Canadian Light Source, a national research facility of the University of Saskatchewan, which is funded by the Canada Foundation for Innovation (CFI), the Natural Sciences and Engineering Research Council (NSERC), the National Research Council (NRC), the Canadian Institutes of Health Research (CIHR), the Government of Saskatchewan, and the University of Saskatchewan.

**Competing interests:** The authors have declared that no competing interests exist.

has become more efficient and more accessible to a wider segment of the population. Unregulated hunting had a disastrous effect on wildlife populations in the late 1800s, including the extinction of the passenger pigeon, and near extinction of the American bison and pronghorn. Conservation efforts including laws and regulations are therefore necessary to prevent catastrophes of the past. In Canada, big game animals are legally considered as a publicly owned natural resource under the jurisdiction of provincial and territorial governments, which set regulations and monitor populations.

Provincial hunter harvest surveys reveal that the most popular method of hunting big game today is with a rifle. Modern rifles can fire a bullet several hundred meters on a very flat trajectory, and with accuracy well below 1 milliradian. There are many different bullet designs, all tailored for different applications [1]. Current laws for all 3 Canadian territories and 7 of the 9 provinces that have big game hunts mandate that the bullets used for big game hunting must be of a type that expands on impact [2]. The expansion process increases the frontal area of the bullet, resulting in increased energy transfer, ensuring the most humane harvest possible. Traditional expanding bullet designs for hunting consist of a lead core with an exposed tip, inside a "jacket" of copper or gilding metal (~95% copper / 5% zinc). Manufacturers often claim the jacket is "bonded" to the core, by processes including swaging [3] and electroplating [4]. Lead is historically used for its combination of density, resulting in high energy transfer, and low cost. The jacket allows these bullets to achieve more than twice the velocity of all-lead designs, which decreases ballistic compensations for wind and distance. It is important to note that there are other established bullet designs, which expand and are legal for hunting, but they do not contain lead [5]. Such designs are commercially available from all leading ammunition manufacturers in a wide range of factory loaded cartridges, including the most popular big game calibers [6].

All projectiles from firearms can be expected to undergo some level of fragmentation while passing through tissues, even if they are not explicitly designed to fragment [1]. This has been known since the late 1800s, but the fragmentation process has attracted greater attention over the past 30 years due to the presence of lead in bullets. While certain metals including iron and copper are essential for life in small amounts, lead can accumulate in tissues [7] but has no known necessary role in biological processes [8]. On the contrary, the toxic effects of lead on human health and the environment have been documented since antiquity [8, 9]. Multinational efforts have legislated the removal and replacement of lead in products, with gasoline, paint, and drinking water distribution systems being among the most well-known examples [10]. As a result of these efforts, the mean blood lead concentration in the USA dropped from 12.8 µg/dL in 1976, to 0.82 µg/dL in 2016, and it continues to fall [10]. In the USA and Canada, 5 µg/dL and 10 µg/dL respectively are now considered levels of concern in adults, whereas in 1971, 60 µg/dL was considered elevated [10].

Researchers have used medical X-ray imaging methods, including radiography, fluoroscopy, and computed tomography, to reveal the presence of millimeter and sub-millimeter-sized fragments along the bullet pathway or "wound channel" of animals harvested with traditional lead-core bullets [11–26]. The vast majority of the fragments observed were considered too small to be sensed while eating. Excision and analysis of these objects confirm the presence of lead [15, 17, 18, 20, 22, 25]. Bullet fragments have also been observed in packages of ground venison, butchered commercially and by hunters themselves [17, 20, 22, 27, 28]. Pigs fed ground venison containing lead bullet fragments displayed a corresponding increase in blood lead levels [17]. Every step of the hypothesized lead exposure pathway, from bullet fragmentation to ingestion to absorption has now been experimentally demonstrated.

We are not aware of any reports which have linked acute lead poisoning in humans with eating wild game hunted with lead-core rifle ammunition. But numerous reports link the

activity with elevated blood lead concentrations [29]. One of the largest studies ($N$ = 736) was performed in North Dakota, USA. Although no participant had a concentration that reached the level of concern at the time of the study, a statistically significant difference in mean blood lead concentrations was found between participants who reported regularly consuming wild game (1.27 μg/dL), and those who did not (0.84 μg/dL) [30]. One extreme example exists of an individual with a blood lead concentration of 74.7 μg/dL. The 54 year old man reported eating exclusively meat, self-harvested with traditional lead-core bullets, for 3 years [27].

Often what remains of a big game animal after butchering, including the internal organs and any damaged meat around the wound channel, is returned to the land. Some animals may not be recovered for a variety of reasons. In a matter of hours to days these remains will be consumed by scavengers. Numerous studies worldwide have linked elevated blood lead concentrations, acute lead poisoning, and deaths in scavengers with ingestion of lead bullet fragments [29]. Wayland and Bollinger investigated lead levels in bald eagle and golden eagle carcasses collected by Canadian provincial wildlife agencies in Saskatchewan, Alberta and Manitoba. Of 131 birds examined, and excluding those that died from being shot, 12% were lethally exposed and a further 4% had elevated lead concentrations, which the authors linked to ingesting lead contaminated carrion [31]. Lead bullet fragments have been directly observed in the gastrointestinal tracts of white-tailed sea eagles that died of lead poisoning [32]. The harmful effects of ingesting lead bullet fragments or lead shot have been reported for more than 100 species of wildlife worldwide [29, 33].

The fragmentation process of projectiles from firearms is important to understand, as fragment composition, size, amount, and distribution all affect toxicity. Fragments were directly observed using lab-based medical X-ray instruments in previous studies [11–28, 32, 34, 35]. Here we apply synchrotron X-ray imaging technology to shine a new light on this process and reveal new details. Synchrotron X-rays are monochromatic and tunable. By making images at photon energies below and above an inner shell electron absorption edge, and then subtracting them, the resulting "K edge subtracted" (KES) images have elemental and, in some cases, chemical contrast [36–38]. Fragment elemental composition can be determined and quantified in an image *a priori*. In addition, the small source size and high brightness of synchrotron radiation improves the spatial resolution of X-ray imaging, well below previous reports on this topic. In this report, we fired bullets of two different compositions into ballistic gelatin with and without embedded bone, to simulate bullet fragmentation in a big game animal. The samples were imaged with lab-based medical radiography and, for the first time, synchrotron radiation from the Canadian Light Source (CLS). The methods are compared, revealing the significance of developments in non-lead bullet designs for big game hunting, and the shortcomings of existing methods to screen wild game meat for lead contamination.

## Materials and methods

Ballistic gelatin blocks with dimensions of 200 mm × 150 mm × 150 mm were purchased from Clear Ballistics LLC. The formulation was "10% gelatin", and the material was calibrated by the manufacturer to meet USA Federal Bureau of Investigation standards, *i.e.*, a 4.5 mm diameter steel ball fired at 180 m/s muzzle velocity should penetrate to a depth between 75 and 95 mm. The blocks were observed to be clear, colorless, and free of defects or foreign material. Intact white-tail deer (*Odocoileus virginianus*) femur bones 25 mm diameter and 150 mm in length were inserted into vertical cuts made in two of four gel blocks, 80 mm from the upstream surface. The bones were obtained from a licensed Saskatchewan resident hunter in the 2020 season, and no animals were killed for the purposed of this research. The CLS health and safety committee reviewed the study protocol under proposal 011660 and approved this research.

Two types of factory loaded 308 Winchester cartridges were purchased locally and used as received: Federal Non Typical Whitetail soft-point (308DT150), which we will refer to as the "lead-core bullet", and Federal Power Shok Copper hollow-point (308150LFA), which we will refer to as the "copper bullet". The advertised bullet weights were the same at 9.72 g (150 grain), as were their muzzle velocities and energies of 860 m/s and 3590 J, respectively. We determined that the lead-core bullets were 74% lead by mass by pulling bullets from cartridges and weighing them before and after heating above 328°C to melt out the lead core. Several copper bullets were also pulled, and these were found to be 3.2 mm longer than the lead-core bullets. The bolt action rifle had a 660 mm long barrel, with 6-groove button rifling and a 1:254 mm right hand twist rate.

Sample preparation was conducted outdoors at an air temperature of -2°C. The gel blocks were transported to the outdoor location in a plastic storage container at 20°C. Blocks were removed from the container, placed on a table, and one shot was fired into each block from a realistic hunting distance of 50 m while a camera recorded the impacts at 960 frames/s. Each block was shot within 30 s of removal from the storage container. The blocks were photographed, and then pure 5 mm × 5 mm foils of 200 μm copper and 165 μm lead were inserted into a corner of each block to act as internal standards for X-ray imaging.

Medical radiographs were acquired using a Carestream DRX Evolution portable digital machine at the Royal University Hospital in Saskatoon, Canada. The instrument contained a tungsten anode X-ray tube, which was operated at 150 kVp with 58.8 mA tube current. No pre-filter was used, and the exposure time for each image was 5.1 ms. The detector was a DRX Plus 3543C, with 2520 × 3032 pixels of 139 μm × 139 μm, and a CsI(Tl) scintillator. The source to detector distance was 1.0 m, and the samples rested directly on the detector during imaging.

Synchrotron imaging was performed at the Biomedical Imaging and Therapy insertion device (BMIT-ID) beamline [39] at the CLS in Saskatoon, Canada. The major components of the beamline include a superconducting wiggler source set to 3.3 T and a double bent Laue Si (111) monochromator. The beamline produces a relatively wide (>100 mm), intense and monochromatic ($\Delta E/E = 5 \times 10^{-3}$) beam, covering a photon energy range of 30–140 keV for advanced X-ray imaging techniques. The photon energy scale of the monochromator was calibrated using metallic foil standards, including a lead foil. The detector system was a tandem indirect X-ray macroscope from Optique Peter. It consisted of an exchangeable scintillator, a 16 bit sCMOS camera (PCO Edge 5.5) with 2560 × 2160 pixels of 6.5 μm × 6.5 μm, and variable optical zoom, providing a tradeoff between spatial resolution and field of view (FOV). For KES measurements, the photon energy was set to 87.00 keV or 89.00 keV, the sample to detector distance was 0.8 m, and the scintillator was 35 μm thick $Gd_2O_2S$:Tb. Optical magnification of 0.15× was used to obtain an effective pixel size of 42.88 μm × 42.88 μm and FOV of 107.20 mm H × 7.76 mm V. Exposure time was 90 ms per image. For high spatial resolution measurements, the photon energy was set to 50.00 keV, the sample to detector distance was reduced to 0.1 m, and the scintillator was changed to 200 μm thick LuAG. The optical magnification was set to 1.8× to obtain an effective pixel size of 3.63 μm × 3.63 μm and FOV of 9.29 mm H × 7.84 mm V. The sample region of interest was thinned from 150 mm to 50 mm to maximize X-ray transmission, and exposure time was increased to 500 ms per image. The source to sample distance was 57.8 m for all measurements at BMIT. Image processing was performed using the program ImageJ [40].

## Results and discussion

### Photographs and video

Photographs comparing the two samples without embedded bone are presented in Fig 1. Viewed from a distance, the wound channels within the samples shot with lead-core bullets

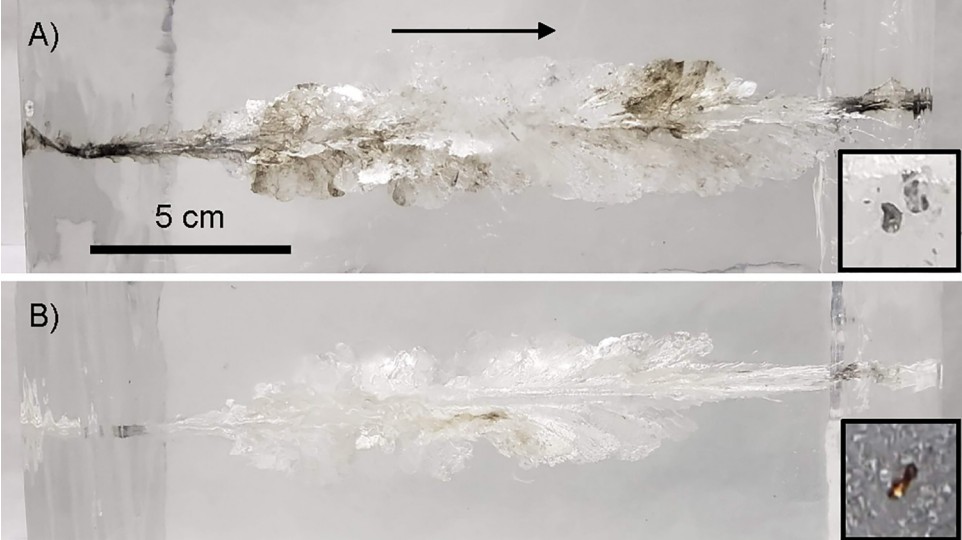

**Fig 1. Photographs of ballistic gelatin block samples.** The bullet trajectory is indicated by the arrow, and both images are on the same spatial scale. The insets at right are examples of embedded bullet fragments clearly visible within the samples. (A) Lead-core bullet without bone. (B) Copper bullet without bone.

(Fig 1A) were obviously darker and cloudier than those shot with copper bullets (Fig 1B). Millimeter-sized metallic fragments were directly observed within the wound channels in three of the four samples. Several dozen embedded metallic fragments were counted within both samples impacted by lead-core bullets, and all of these fragments appeared silver-white or lead-like in color. In comparison, the copper bullet without bone sample contained a single visible fragment, which was copper in color, and the copper bullet with bone sample had no visible metallic fragments that could be clearly distinguished from bone and bone marrow fragments. The dimensions of the wound channel were similar for both bullet types, extending to approximately the same maximum diameter of 45 to 60 mm at 80 to 120 mm depth.

Rifles are manufactured in dozens of calibers, and there are dozens of ammunition manufacturers worldwide, with each producing several types of hunting bullets. Shot distances can range from 10 m to over 500 m. There are several species of big game animals, and each species has several tissue types where the bullet impact could happen. These variables lead to thousands of possible impact outcomes. Fortunately, the literature contains hundreds of data points on hunting bullet impacts, including those from all major ammunition manufacturers, in all common calibers, at a range of distances, in many types of animal tissue and tissue simulants including ballistic gelatin. Three key and generalizable traits have emerged in the literature,

1. Traditional lead-core hunting bullets produce many metallic fragments on impact, also referred to as a "lead snowstorm".

2. Copper hunting bullets produce few to no metallic fragments on impact.

3. The size of the wound channels has been characterized for both bullet types, at typical hunting distances in several animal tissues.

Although our study is based upon four impacts, due to the limited amount of synchrotron beamtime available, our observations of fragmentation and wound channel dimensions are in agreement with previous studies [11–13, 16, 23, 34]. They indicate that these four impacts are representative for the bullet type, and that ballistic gelatin is a fine tissue substitute [1, 35]. The

embedded femur bones shattered on impact. The lead-core bullet with bone sample retained significant bone fragments and marrow, while the copper bullet with bone sample retained relatively few bone fragments. High-speed video revealed a similar violent impact for both bullet types (S1–S4 Files). We observed the lead-core bullet had a direct impact on the bone, while the copper bullet had a glancing impact on the bone and the majority of the bone mass happened to be ejected from the block. For all samples, the main bullet masses exited the blocks and were not recovered. Stopping and capturing the main bullet mass would have likely required more than 400 mm of ballistic gelatin [35].

## Medical radiography

Medical radiographs of all four samples are presented in Fig 2. Medical radiographs are X-ray transmission images collected using a polychromatic hard X-ray source. Contrast in these images is proportional to the X-ray photoabsorption of materials, and is described by the equation,

$$I = I_0 e^{-(\mu/\rho)\rho t} \tag{1}$$

where $I$ is transmitted intensity, $I_0$ is incident intensity, $\mu$ is the linear attenuation coefficient (cm⁻¹), $\rho$ is density (g/cm$^3$), and $t$ is thickness (cm). Values of $\mu/\rho$, also known as the mass attenuation coefficient (cm$^2$/g), have been measured and tabulated for many elements and materials [41]. The mass attenuation coefficients for materials relevant to this study are plotted in Fig 3.

The bullet trajectories were clearly observable as a disruption of the uniform gel. Bone fragments, marrow, and void air space around the bone were visible for the samples with embedded bone. The gel had the appearance of being more absorbing moving away from the center of the image, and the edges and corners also appeared to be slightly blurred. These aspects are

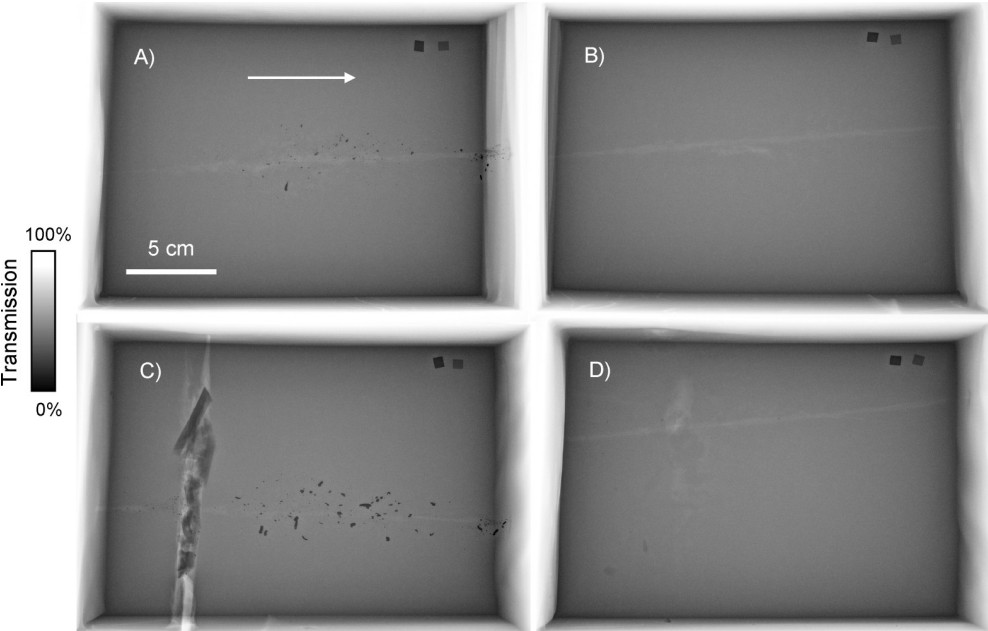

**Fig 2. Medical radiographs of ballistic gelatin block samples.** The bullet trajectory is indicated by the arrow, and all images are on the same spatial and contrast scales. (A) Lead-core bullet without bone. (B) Copper bullet without bone. (C) Lead-core bullet with bone. (D) Copper bullet with bone.

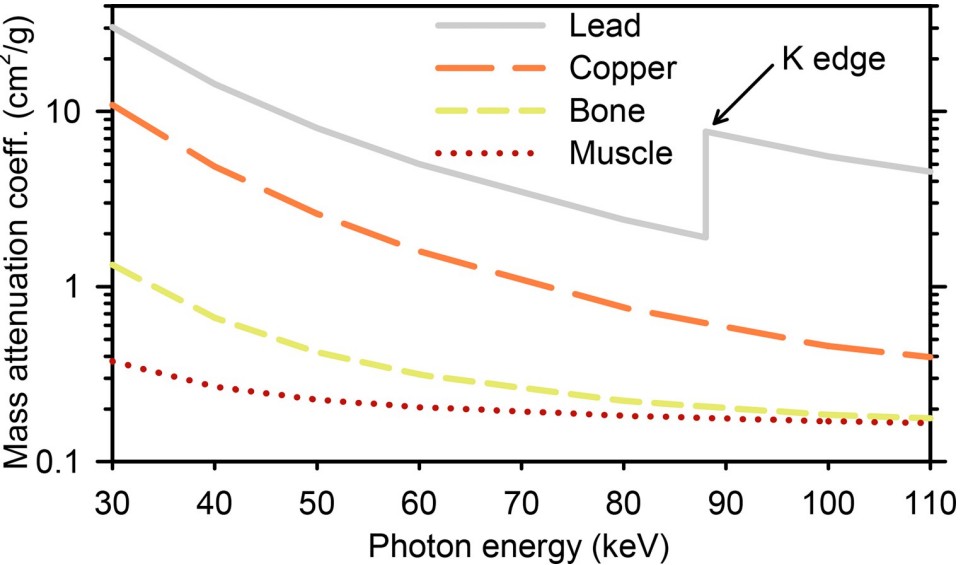

**Fig 3. X-ray mass attenuation coefficients for lead, copper, cortical bone, and skeletal muscle [41].**

geometric artefacts, related to the cone-shaped X-ray emission of the tungsten anode source and the short source-to-detector distance.

All samples contained highly absorbing bullet fragments along the wound channel. The copper bullet samples contained very few fragments (8 observed in the sample without bone, 5 in the sample with bone), while more than 100 fragments could be resolved within each lead-core bullet sample. The copper bullets are advertised as being lead-free, therefore the fragments of those bullets were not expected to contain lead. But the lead-core bullets contain both the lead core and the copper jacket. Fragments from these bullets could be composed of lead, copper, or both metals. Can the medical radiographs be used to determine which fragments are lead and which are copper? The mass attenuation coefficient of lead is at least 2.5× greater than copper across the broad photon energy range of approximately 20 to 150 keV that produced these radiographs [41] (Fig 3). Therefore, a pure lead foil will be significantly less transmissive than a pure copper foil of equal thickness (Eq 1). Consider the metallic foil internal standards which are captured in the upper right corner of all radiographs. We have *a priori* knowledge of the foil purity and thickness, and can therefore state conclusively that the foils on the left of the radiographs are lead and the foils on the right are copper. However, concerning the lead-core bullet samples, we do not have *a priori* knowledge of fragment purity and thickness. Therefore, common medical radiographs do not provide elemental contrast; lead fragments cannot be distinguished from copper fragments.

## Synchrotron KES imaging

The KES imaging method is demonstrated for the lead-core bullet with bone sample in Fig 4. The 1s or K shell electrons of lead can be promoted into unoccupied electronic states at a characteristic photon energy of 88.0045 keV or greater [41] (Fig 3). Monochromatic transmission X-ray images of each block were collected at photon energies slightly below the lead K edge (87.00 keV, Fig 4A), and slightly above it (89.00 keV, Fig 4B). The final quantitative KES image is presented in Fig 4C, and the procedure to create it is described in S5 File. The internal standard foils demonstrate the power of KES imaging in comparison to medical radiography. The transmission of the lead foil decreased by 51% in the image at 89 keV (Fig 4B) relative to the

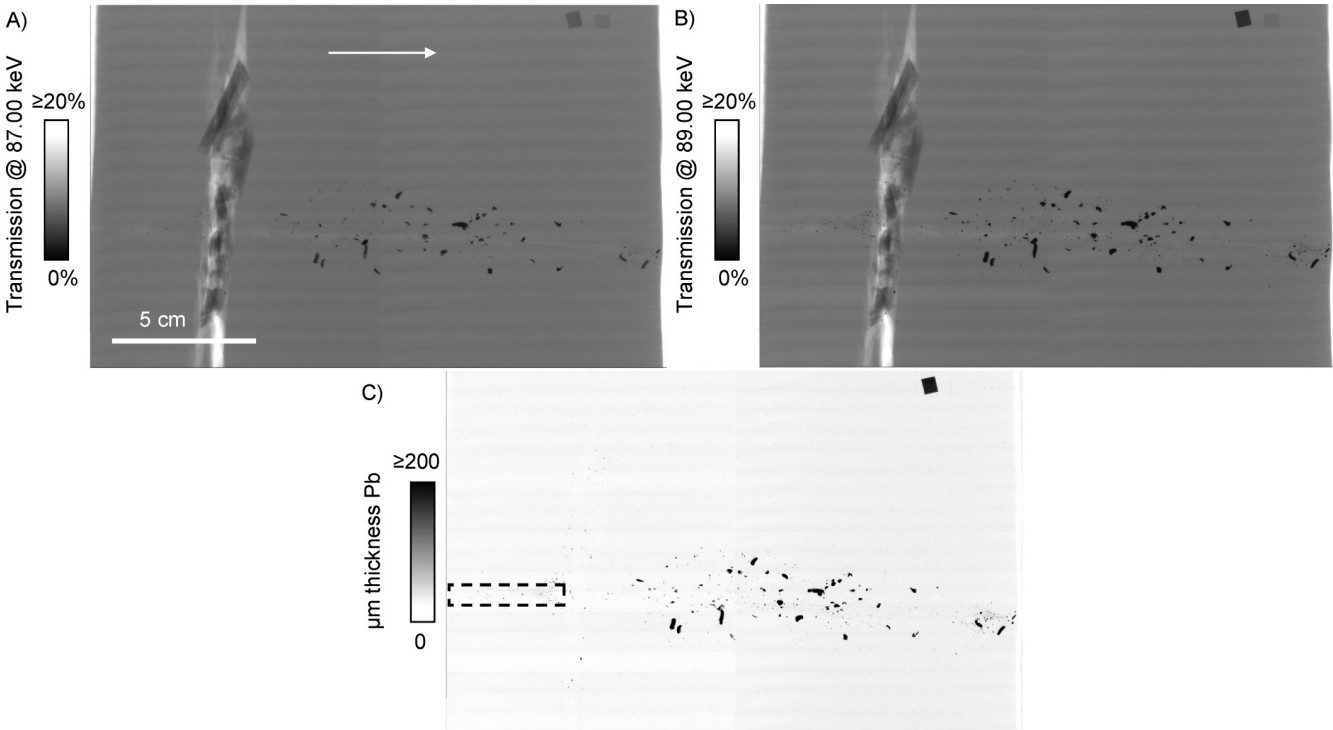

**Fig 4. Lead K edge subtraction imaging of the lead-core bullet with bone sample.** The bullet trajectory is indicated by the arrow, and all images are on the same spatial scale. (A) Transmission image at 87.00 keV. (B) Transmission image at 89.00 keV. (A) and (B) are on the same contrast scale. (C) K edge subtraction image, with contrast scale of micrometer thickness of lead. All non-lead structures including bone and copper are subtracted out in the process. The dashed box in (C) outlines an area that was selected for high resolution imaging.

image at 87 keV (Fig 4A), while the transmission of the copper foil at both energies was within 2%. Copper and other non-lead structures such as bone fragments do not experience significant additional absorption in this relatively small energy range (Fig 3). When the 87 keV image is subtracted from the 89 keV image, all non-lead structures including the copper foil disappear. The remaining contrast in the KES image is now specific and quantitative for lead [36–38].

The KES images of the copper bullet samples (not shown) had no significant contrast except for the lead internal standard foil. All of the components in the copper bullet samples (bullet fragments, bone, gel) were anticipated to contain negligible levels of lead, and this result confirms the expectation. For the lead-core bullet samples, essentially all of the bullet fragments observed in the transmission images remained in the KES images. It was difficult to locate a single fragment, large or small, which completely disappeared in the KES image in a similar manner to the copper foil internal standard, indicating that practically all bullet fragments in the lead-core bullet samples contained lead.

Resolving what is and what is not a bullet fragment amongst the remains of a harvested animal using medical radiography relies on operator judgement. The typical internal structure may be drastically altered by the bullet impact, bone fragments may be widely dispersed, and wild animals can contain a variety of foreign matter [17, 21, 22]. Furthermore, if the bullet is not homogenous, the bullet fragments may have different compositions. Fragment composition is critical for further discussions of toxicity: Lead fragments are toxic, whereas copper fragments have been shown to be non-toxic to birds and mammals including humans in the amounts produced by copper bullets [6]. Some researchers in this field recognize that medical

radiographs are not selective for lead [11, 21, 23, 25, 27], yet in many cases all suspect fragments in a radiograph were counted as lead [11, 14, 15, 19, 20, 21]. Medical radiography, the most common technique applied in bullet fragmentation studies, is simply not a conclusive measurement of elemental composition.

Distinction of lead fragments from copper and other material has been accomplished by spectrometric and spectroscopic techniques. Previous examples involve targeted extraction, dissolving and analyzing a select number of fragments [17, 20], dissolving large volumes of tissue suspected to contain fragments and analyzing the bulk liquid [12, 15, 18, 42], or analyzing fragments separated from the supernatant [22, 25, 28]. Some of these examples reveal the shortcomings of medical radiography. Hunt *et al.* used inductively coupled plasma-mass spectrometry (ICP-MS) to analyze radio-dense fragments excised from packages of ground venison. Lead was confirmed in 93% of the fragments, and the remaining 7% discrepancy was attributed to copper [17]. Pauli and Buskirk analyzed metallic fragments from hunted prairie dogs using flame atomic absorption spectroscopy, and 28% of the mean mass of fragments from expanding lead-core bullets was found to be copper [25]. Officials from the North Dakota Departments of Health and Agriculture screened 404 samples of ground venison for lead bullet fragments using medical radiography, and identified 49 samples with suspicious objects. These underwent ICP-MS analysis, which confirmed that lead was present in only half [24] of the samples. The suspicious objects in the other half were found to be "bone fragment[s], plastic or other type of metal, such as copper" [28]. The bullet fragmentation process has many variables, and the literature examples show that fragment composition cannot be assumed.

In this report, we demonstrate how KES is a superior form of X-ray imaging for visualizing bullet fragmentation. The technique combines imaging and X-ray absorption spectroscopy, thereby removing ambiguity and operator judgement involved in analyzing medical radiographs. KES imaging has the ability to resolve only a single element of interest, finely divided and embedded within a large and non-homogeneous sample volume. When large volumes of tissues are dissolved for elemental analysis, information about individual particles and their distribution is lost. When individual fragments are targeted and removed for elemental analysis, the results may be skewed from preferential sampling. We found targeted excision is increasingly challenging for small fragments, even with our optically transparent samples. The KES technique provides an elemental concentration map with high spatial resolution. KES imaging for other elements including copper is possible, but was not attempted as the copper K edge (8.97 keV) does not fall within the accessible photon energy range of the BMIT-ID beamline used for this investigation. Absorption saturation would also become a greater issue as X-ray attenuation lengths for all materials decrease with decreasing photon energy [41], so samples may need to be thinned.

## Synchrotron high resolution imaging

Several dark and cloudy regions were visible along the wound channels of the lead-core samples (Fig 1A). Radiographs and KES images of these regions revealed many small lead fragments. One such region, outlined in Fig 4C, was selected for high resolution imaging at an effective pixel size of 3.63 μm × 3.63 μm. Images of six sub-regions were collected and stitched together to form a 41.30 mm × 7.85 mm transmission image (Fig 5A). ImageJ [40] was used to threshold the transmission image, measure fragment diameters, and create a histogram of fragment diameter (Fig 5B). Thousands of fragments were resolved in this small area, including many with diameters of 2 or 3 pixels (7 to 11 μm) (Fig 5C). Of the 12784 fragments counted just in this region, 8152 had diameters <20 μm. Only 115 fragments had diameters >140 μm. The remaining 12669 fragments with diameters <140 μm would likely not be resolvable using

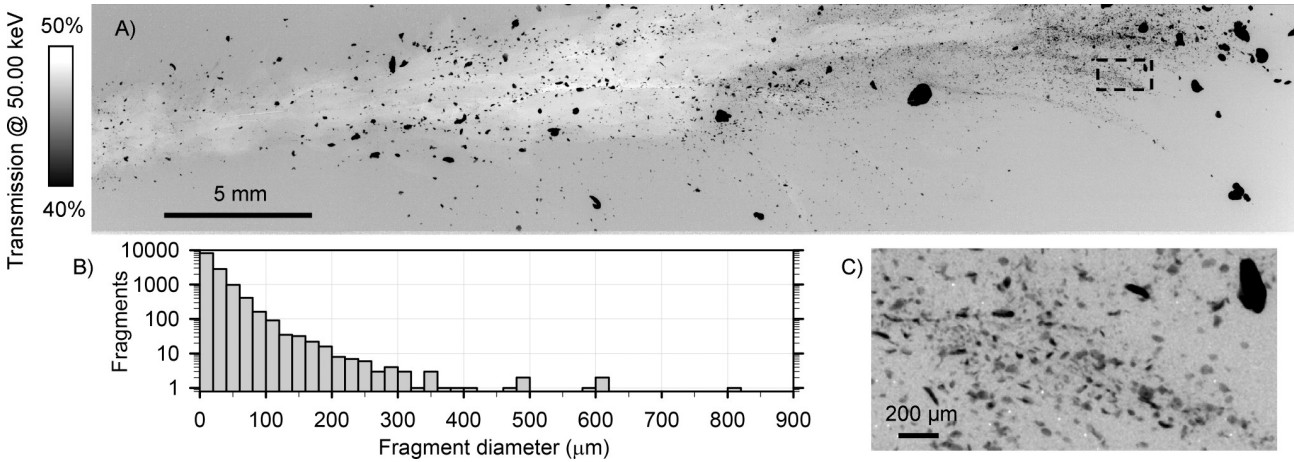

**Fig 5.** (A) High resolution X-ray transmission image of a select region of the lead-core bullet with bone sample. (B) Histogram of fragment diameters, derived from (A). (C) Zoomed in region, 2 mm × 1 mm, indicated by the dashed box in (A), revealing thousands of sub-100 μm diameter fragments.

common medical X-ray imaging instruments. For reference, the 200 μm scale bar in Fig 5C is comparable to the 139 μm pixel size of the detector used to collect the medical radiographs (Fig 2).

Finely divided lead particles have proven to be more bioavailable than larger particles. A fivefold enhancement of absorption of lead was observed in rats for 6 μm mean diameter particles compared to 197 μm particles [43]. In addition, Krone *et al*. have demonstrated that white-tailed sea eagles will avoid large metal fragments dispersed within carcasses while consuming the smaller ones [23]. Although smaller fragments play a greater role when evaluating toxicity, fragment size has received little attention in previous bullet fragmentation studies. Medical radiography has been the main and often exclusive technique employed to visualize embedded bullet fragments, yet surprisingly, the majority of previous studies do not report the minimum spatial resolution achieved [12, 14, 16–22, 24–28, 35]. Where minimum spatial resolution values were reported [11, 13, 15, 26, 34], they were often estimates of unclear origin and, more importantly, not better than unaided human vision (>0.1 mm). To our knowledge, our report contains the first application of X-ray microscopy (<0.1 mm) to bullet fragmentation processes. We have directly observed embedded lead fragments in the 100 μm to single digit micrometer diameter size regime where they were not previously known to exist. The spatial resolution in our investigation was ultimately limited by the effective pixel size of the detector, and it is likely that smaller particles could be resolved at higher spatial resolution. Kollander *et al*. detected lead nanoparticles (40–750 nm diameter) near the wound channels in hunted roe deer and wild boar using mass spectrometry [15]. However, metallic lead is known to dissolve in tissue over time [44], and indeed the authors noted a significant background signal originating from dissolved lead [15]. The wet chemical sample preparation necessary for the technique may have also affected the particle size distribution. X-ray microscopy is a more direct technique for answering these questions, as it typically requires little to no sample preparation/alteration, and can accommodate thick, hydrated samples. There are several hard X-ray nanoprobe beamlines at synchrotrons which offer routine spatial resolutions below 100 nm.

The existence of thousands of single digit micrometer-sized particles, comparable to mammalian erythrocytes [45], challenges the current understanding of the maximum extent that fragments may be distributed in a harvested big game animal. Though it has been shown that lead concentrations are highest in the immediate vicinity of the wound channel [42], bullet

fragments have been previously observed up to 45 cm from the bullet pathway [17]. Typically, a big game animal is not instantly incapacitated, and during its final moments, micrometer-sized fragments could feasibly enter the bloodstream and be transported throughout its body. Ingestion of very small lead particles would explain the accumulation of lethal and sub-lethal lead levels in scavengers in cases where no fragments were found in a medical radiograph or biopsy [11, 31]. Some American states including Minnesota [46] and Wisconsin [22] have screened wild game meat donated to food banks for lead fragments using medical radiography. Our results demonstrate that 1) medical radiography is not selective for lead, and 2) common medical X-ray instruments do not achieve the spatial resolution required to resolve the smallest and most bioavailable lead fragments. It is possible for a package of venison containing thousands of sub-100 μm lead fragments to pass this screening test, while a package with one radio-dense object would fail the screening and that object may not contain any lead.

## Alternatives to traditional rifle ammunition

We demonstrated that both lead-core and copper bullets had a similar devastating effect on ballistic gelatin. We also found the accuracy was equivalent: Five consecutive shots at 100 m were within a 40 mm diameter, and no adjustments to the rifle were required when switching between the two types. The cartridges used in this study were stocked locally, and the price of the copper bullet option chosen for this study ($41 CDN) fell between the budget ($32 CDN) and premium ($55 CDN) 308 Winchester ammunition options locally available. These observations are in agreement with previous studies comparing the practicality and effectiveness of copper bullets for big game hunting [6, 47, 48]. Copper is not as dense as lead, normally requiring the mass equivalent copper bullets to be slightly longer, which can affect accuracy [6]. Trials are the only way to determine the accuracy potential of any bullet for an individual rifle.

Sustainably hunted wild game is often excellent in quality, low in fat, high in nutritional value, and critical to preservation of culture. Non-lead rifle ammunition maintains all of those benefits, while avoiding the known negative effects of lead on human health and the environment. Some regions, including the countries of Sweden, Denmark, and the American state of California, have mandated the use of non-lead rifle ammunition for hunting. Nearly all Canadian provinces and territories already have laws regarding the types of bullets that can be used for hunting big game. These laws are rooted in ethical concerns, for example, bans on the use of non-expanding bullets, and certain calibers considered not powerful enough to be effective [2]. We are not aware of any legislation prohibiting the use of lead rifle ammunition for hunting anywhere in Canada. However, the 2021–22 hunting regulation handbooks published by the provinces of Nova Scotia and Prince Edward Island contain statements on the health and environmental hazards of lead rifle ammunition [49, 50]. We recommend that all Canadian provinces and territories encourage the use of non-lead ammunition for hunting in their annual hunting regulation handbooks. Hunting with non-lead ammunition is the only 100% effective method to avoid the proven health and environmental hazards of lead projectile fragments dispersed within wild game meat.

## Supporting information

**S1 File. High-speed video of a lead-core bullet impacting a ballistic gelatin block.**
(MP4)

**S2 File. High-speed video of a lead-core bullet impacting a ballistic gelatin block containing embedded bone.**
(MP4)

**S3 File. High-speed video of a copper bullet impacting a ballistic gelatin block.**
(MP4)

**S4 File. High-speed video of a copper bullet impacting a ballistic gelatin block containing embedded bone.**
(MP4)

**S5 File. Procedure to create K edge subtracted (KES) images.**
(DOCX)

## Acknowledgments

We thank Jason Shambel, Terry Wawruck and Glenn Leontowich for assistance with sample preparation, and David Muir for performing the X-ray fluorescence measurements.

## Author Contributions

**Conceptualization:** Adam F. G. Leontowich.

**Data curation:** Adam F. G. Leontowich, Arash Panahifar, Ryan Ostrowski.

**Formal analysis:** Adam F. G. Leontowich.

**Investigation:** Adam F. G. Leontowich, Arash Panahifar.

**Methodology:** Adam F. G. Leontowich.

**Project administration:** Adam F. G. Leontowich.

**Resources:** Adam F. G. Leontowich.

**Supervision:** Adam F. G. Leontowich.

**Visualization:** Adam F. G. Leontowich.

**Writing – original draft:** Adam F. G. Leontowich.

**Writing – review & editing:** Adam F. G. Leontowich, Arash Panahifar, Ryan Ostrowski.

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
