## [Decision Letter · Decision Letter 0]

17 May 2022

PONE-D-22-05415Fragmentation of hunting bullets observed with synchrotron radiation: Lighting up the source of a lesser-known lead exposure pathwayPLOS ONE

Dear Dr. Leontowich,

Thank you for submitting your manuscript to PLOS ONE. After careful consideration, we feel that it has merit but would be improved by implementing the comments from the two, generally favorable, reviewers. In addition, implementing these changes will allow your manuscript to better meet PLOS ONE’s publication criteria. Therefore, we invite you to submit a revised version of the manuscript that addresses the points raised during the review process.

Please specifically focus on the comments from Reviewer #1 which request expansion of the discussion of the limitations of the study. Also, please note an important potential correction identified by Reviewer #2 in equation 1. 

We look forward to receiving your revised manuscript.

Kind regards,

Christopher James Johnson, Ph.D.

Section Editor

PLOS ONE

Journal Requirements:

“Part of the research described in this paper was performed at the Canadian Light Source, a national research facility of the University of Saskatchewan, which is funded by the Canada Foundation for Innovation (CFI), the Natural Sciences and Engineering Research Council (NSERC), the National Research Council (NRC), the Canadian Institutes of Health Research (CIHR), the Government of Saskatchewan, and the University of Saskatchewan.”

“The authors received no specific funding for this work.”

Reviewers' comments:

Reviewer's Responses to Questions

**Comments to the Author**

1. Is the manuscript technically sound, and do the data support the conclusions?

Reviewer #1: Yes

Reviewer #2: Yes

2. Has the statistical analysis been performed appropriately and rigorously? 

Reviewer #1: N/A

Reviewer #2: Yes

3. Have the authors made all data underlying the findings in their manuscript fully available?

Reviewer #1: Yes

Reviewer #2: Yes

4. Is the manuscript presented in an intelligible fashion and written in standard English?

Reviewer #1: Yes

Reviewer #2: Yes

5. Review Comments to the Author

Reviewer #1: The authors used synchrotron K edge subtraction imaging to investigate the fragmentation of a traditional lead-core bullet after impact on ballistic gelatin and compared it with the fragmentation of a copper bullet. The synchrotron K edge subtraction imaging method was applied the first time on this subject and allowed to image specifically the lead fragments. A statistical analysis of the size distribution of the lead particles down to particle sizes of a few micrometer showed for the first time that the vast majority of particles is smaller than detectable by visual inspection.

The study is very clearly and carefully written, technically sound and the topic is of relevance for human health and environment.

However, the study has some limitations in my view, which the authors should comment on:

1. The study is based only on four impacts: two with a lead-core bullet (with and without bone), and two with a copper bullet (with and without bone).

a. It is not clear whether the results of this particular lead-core bullet can be generalized to all or other popular types of lead-core bullets.

b. It is not clear whether this single impact is representative for this bullet type.

c. It is not clear whether the fragmentation is the same for ballistic gelatin, which is used in the study, and for a real big game animal.

2. The synchrotron K edge subtraction imaging method is standard at synchrotrons. The novelty of this study lies therefore exclusively in its application. Would the method also work with similar sensitivity on real big game animals? Please comment on expected limitations.

3. What is the market share of lead-core bullets compared to nonlead-core bullets today? Are lead-core bullets still relevant?

4. In line 346 ff you reason about the maximum extent that fragments may be distributed in a harvested big game animal. I suggest to use your data to compile a size dependent radial distribution of the fragments (of cause you only have the measurement of the projection of the distribution, estimate the effect).

In conclusion, the publication of the study is recommended. The comments should be considered. In the best case it helps to expulse lead-core bullets worldwide. Whether PLoS ONE is the appropriate journal is to be assessed by the editor.

Reviewer #2: In the paper, KES imaging is used to identify lead bullet fragments in animal remains among other metal (copper) and bone fragments. High resolution capabilities of the technique is also evaluated. A comparison with the widely diffused medical radiography imaging shows the advantages of the synchrotron KES imaging.

Material and methods are well discussed and results fully reproducible. Bibliography is up-to-date and pertinent. Text is clear, logical and well written.

Just a few remarks: in eq.1 line 206, “t” where (line 208) “t is thickness (cm)” makes no sense. It should be lambda, i.e. mass thickness in g/cm2.

I would move line from 266 to 292 in the Introduction (it’s basically bibliography) The same for lines from 324 to 333 and from 338 to 345.

ImageJ software needs to be described in the Methods.

Is the last paragraph “Alternatives to traditional rifle ammunition” really required in a research paper like this? I feel it’s redundant.

6. PLOS authors have the option to publish the peer review history of their article (what does this mean?). If published, this will include your full peer review and any attached files.

Reviewer #1: No

Reviewer #2: **Yes: **Matteo Donghi

---

## [Author Response · Author response to Decision Letter 0]

20 Jun 2022

Thank you for considering our revised manuscript!

Funding information has been removed from the acknowledgements section as requested. The information removed was related to funding that the CLS facility receives, that all users of CLS are requested to cite. 

Our funding statement remains the same: "The authors received no specific funding for this work".

---

## [Editor Report · Decision Letter 1]

12 Jul 2022

Fragmentation of hunting bullets observed with synchrotron radiation: Lighting up the source of a lesser-known lead exposure pathway

PONE-D-22-05415R1

Dear Dr. Leontowich,

We’re pleased to inform you that your manuscript has been judged scientifically suitable for publication and will be formally accepted for publication once it meets all outstanding technical requirements.

Kind regards,

Christopher James Johnson, Ph.D.

Section Editor

PLOS ONE
---

## [Editor Report · Acceptance letter]

2 Aug 2022

PONE-D-22-05415R1 

Fragmentation of hunting bullets observed with synchrotron radiation: Lighting up the source of a lesser-known lead exposure pathway 

Dear Dr. Leontowich:

I'm pleased to inform you that your manuscript has been deemed suitable for publication in PLOS ONE. Congratulations! Your manuscript is now with our production department. 

Kind regards, 

on behalf of

Dr. Christopher James Johnson 

Section Editor

PLOS ONE